# 3,3′-Diindolylmethane Improves the Viral Pneumonia Outcomes After Influenza and SARS-CoV-2 Infection in Animal Models

**DOI:** 10.3390/v17070964

**Published:** 2025-07-09

**Authors:** Vsevolod Kiselev, Irina Leneva, Anna Ivanina, Artem Poromov, Irina Falynskova, Nadezhda Kartashova, Ekaterina Glubokova, Galina Trunova, Sergey Sudakov, Vadim Drukh, Vitaly Zverev, Oleg Kiselev

**Affiliations:** 1Academician V.I. Kulakov National Medical Research Center for Obstetrics, Gynecology, and Perinatology, Ministry of Health of Russia, 117997 Moscow, Russia; vkis10@mail.ru (V.K.);; 2Mechnikov Research Institute for Vaccines and Sera, 105064 Moscow, Russia; ivanina.anna97@mail.ru (A.I.); poromov@insmech.ru (A.P.); falynskova@mail.ru (I.F.); nadezdakartasova10571@gmail.com (N.K.); eaglubokova@yandex.ru (E.G.); vitalyzverev@outlook.com (V.Z.); 3P.A. Hertsen Moscow Oncology Research Institute, 125284 Moscow, Russia; gtrunovamnioi@mail.ru; 4AO MiraxBioPharma, 121059 Moscow, Russia; ssudacov77@gmail.com (S.S.); druhvm@gmail.com (V.D.)

**Keywords:** 3,3′-Diindolylmethane, DIM Epi, DIM, oseltamivir, molnupiravir, antiviral combination, influenza, SARS-CoV-2, viral pneumonia

## Abstract

Influenza and SARS-CoV-2 are often associated with viral pneumonia, resulting from direct exposure of the virus to lung tissue. 3,3′-Diindolylmethane (DIM) is a naturally occurring substance with multi-target activity, including anti-inflammatory and epigenetic modulation. In this study, we evaluated the therapeutic efficacy in vivo of a DIM formulation with fish oil (Cesarox Epi) against influenza A (H1N1) infection in mice and against SARS-CoV-2 infection in Syrian hamsters. In a model of lethal influenza pneumonia induced by A/California/04/2009 (H1N1)pdm09 virus, we showed that 5 days’ treatment with DIM Epi at 10, 20, and 60 mg/kg/day delayed the time to death, prevented body weight loss, and resulted in significant improvements in survival. DIM Epi tested in hamsters infected with SARS-CoV2 Dubrovka (Wuhan-like) strain at doses 50 and 100 mg/kg/day reduced clinical signs, weight loss, temperature elevation, and lung pathology. In both models of infections, treatment with DIM Epi did not significantly decrease viral titer in the animals’ lungs. DIM Epi and Oseltamivir were more effective against influenza infection when given in combination than given singly, while co-administration of DIM Epi with Molnupiravir did not yield an additive benefit against SARS-CoV-2 infection. These findings support DIM Epi as a promising host-directed adjunct therapy for viral pneumonia with potential to enhance outcomes in respiratory infections.

## 1. Introduction

The coronavirus 2019 disease (COVID-19) pandemic, caused by severe acute respiratory syndrome coronavirus 2 (SARS-CoV-2), has had a substantial and multifaceted impact on global public health over the past 4 years. As of 11 May 2025, the World Health Organization (WHO) has recorded 778 million confirmed cases and approximately 7.1 million cumulative fatalities [1]. The disease is caused by severe acute respiratory syndrome coronavirus 2 (SARS-CoV-2) and can manifest as mild, severe, or critical illness. Critical disease is frequently associated with viral pneumonia and acute respiratory distress syndrome (ARDS) with respiratory and multi organ dysfunction [2,3]. Following a rigorous campaign involving measures such as vaccinations, medications, and restrictions on social activities, the COVID-19 pandemic has been managed under control. On 5 May 2023, the WHO downgraded the global emergency status of COVID-19 [4]; however, this declaration does not imply that the fight against infectious diseases has concluded. The pandemic has accelerated scientific research aimed at identifying effective therapeutic interventions to combat COVID-19. Several direct-acting antivirals against SARS-CoV-2 have received approval or emergency use authorization, but all of them have limitations, and further study to confirm their effectiveness is required [5].

Another major respiratory threat, seasonal influenza viruses account for approximately one billion cases each year, with 290,000 to 650,000 deaths globally [6]. The co-circulation of influenza and SARS-CoV-2 has created a dual threat to global health systems [7]. This scenario is complicated further by the potential pandemic threat posed by zoonotic influenza viruses. Human infections with avian and other zoonotic influenza viruses are relatively rare but are associated with higher lethality rates due to the absence of pre-existing immunity in the population [8,9]. Influenza virus can also cause severe illness (such as sepsis, septic shock, severe pneumonia, acute respiratory distress syndrome [ARDS], multi-organ failure, exacerbation of chronic medical conditions) or death [10]. Current FDA-approved therapeutics have a narrow treatment window. In addition, the influenza virus can overcome the effects of the antiviral agents and develop drug resistance [11].

Both influenza and SARS-CoV-2 infections are often associated with viral pneumonia, which results from direct exposure of the virus to lung tissue. A key factor in the pathogenesis of severe viral pneumonias, including influenza and coronavirus infections, is the excessive and prolonged secretion of pro-inflammatory cytokines. While macrophages are major contributors to this process, other immune cells such as neutrophils, dendritic cells, and T cells, as well as damaged epithelial and endothelial cells, also participate in the production of inflammatory mediators. The development of ARDS is multifactorial, resulting from a combination of direct viral cytopathic effects and the host’s individual immune response. This hyper-inflammatory state can lead to widespread lung inflammation, respiratory failure, and death [12].

Preventing and controlling influenza and SARS-CoV-2 infections remains a global public health challenge, as it causes seasonal epidemics to unexpected pandemics. Apart from developing new classes of direct-acting antiviral drugs that act on viral factors, targeting the essential host factors needed for viral entry, replication, and release is another important area for therapeutic interventions. The study of the interaction between the viruses and various host proteins is another important area of active investigation in recent years [13]. Targeting host factors offers several advantages, including a lower risk of development of resistance, as most of these factors are evolutionarily conserved. Additionally, this approach has potential to be effective against multiple respiratory viruses, as they often share common molecular pathways. One of the approaches is drug repurposing, which has become increasingly popular to facilitate all stages of drug discovery and development, as it has the potential to reduce time and lower costs. During the COVID-19 pandemic, drug repurposing has been an effective strategy to rapidly respond to medical emergencies. Another approach to improve the therapy of viral infections is different combinations for enhanced potency and reduced emergence of drug-resistant viruses. It is hypothesized that combinations of antiviral agents targeting different pathways may be particularly beneficial for treating severe forms of infection [14,15]. Given the above considerations, it is logical to assume that antiviral therapy should be accompanied by a host of ranged agents because antiviral drugs do not directly inhibit the inflammatory processes associated with pulmonary injury.

3,3′-Diindolylmethane (DIM, Figure 1), a compound found in cruciferous vegetables, exhibits potent anti-inflammatory and anticancer properties. DIM suppresses T cell activation induced by staphylococcal enterotoxin B through epigenetic regulation involving histone deacetylase (HDAC) expression [16]. In murine macrophages, DIM inhibits lipopolysaccharide-induced release of pro-inflammatory mediators by downregulating NF-κB and AP-1 signaling pathways [17]. DIM attenuates colonic inflammation and tumorigenesis in mouse models of colitis and colitis-associated colorectal cancer [18]. Mechanistically, DIM acts as a mitochondrial H+-ATP synthase inhibitor, leading to increased reactive oxygen species (ROS) production and subsequent activation of stress-activated pathways involving p38 and JNK. This oxidative stress-induced signaling cascade results in p21 overexpression, contributing to DIM anticancer effects in human breast cancer cells [19]. These studies highlight DIM’s potential as a therapeutic agent for cancer prevention and inflammatory conditions potentially including cytokine storm developing during viral pneumonia.

The only major obstacle to creating an effective DIM-based antiviral drug is the low bioavailability of the crystalline form of this substance. DIM shows low solubility in physiological liquids and has very limited ability to penetrate through barrier membranes. Cesarox^®^ Epi is the only known formulation of DIM, developed by LLC Epipharma (Moscow, Russia), that provides high bioavailability and maximum therapeutic efficacy of DIM. Each capsule contains 100 mg of DIM, 20 mg of type A fish oil, and a polysorbate (ratio 15:2:58). This composition was developed to enhance the solubility and oral bioavailability of DIM, as described in Russian patent RU 2,419,426 C1 (published on 27 May 2011) [20]. All previously conducted studies described above were performed using the DIM substance alone. However, the composition combining fish oil and DIM has been less studied. Due to its improved pharmacokinetic properties, this formulation may offer greater therapeutic efficacy in the treatment of viral infections as part of combination therapy.

Experimental studies [20] in a mouse lethal model of influenza pneumonia induced by H3N2 virus demonstrated that DIM Epi -based formulations exhibited high protective activity when administered orally. The formulations significantly reduced mortality rates in experimental groups compared to controls and increased the average survival time, indicating a level of efficacy comparable to or exceeding that of the reference drug, rimantadine. Despite these promising outcomes, the study revealed no impact of the formulation on influenza virus replication in the lungs of infected animals. However, DIM Epi formulations effectively prevented chronic post-influenza complications, significantly reducing the extent of chronic post-influenza pneumonia lesions.

In the present study, we aimed to evaluate the universal therapeutic potential of Cesarox Epi (hereafter DIM Epi) as a multi-target agent in viral pneumonia caused by respiratory pathogens. To this end, we investigated its efficacy, alone and in combination with direct-acting antivirals, in two distinct animal models of severe respiratory infections: influenza A (H1N1)pdm09 in mice and SARS-CoV-2 in Syrian hamsters.

## 2. Materials and Methods

Viruses and Cells. Influenza A/California/04/2009 (H1N1)pdm09 virus was provided by the WHO National Influenza Centre of Russia (St. Petersburg, Russia) and mouse-adapted by three lung-to-lung passages. The influenza A/Aichi/2/69 (H3N2) virus was from the collection of the Federal Research Centre of Epidemiology and Microbiology named after Honorary Academician N. F. Gamaleya (Moscow, Russia). The virus was grown in the allantoic cavity of 9-day-old embryonated chicken eggs for 48 h at 37 °C.

SARS-CoV-2 virus (‘Dubrovka’ strain, GenBank: MW 161,041.1) was isolated from the nasopharynx aspirate and throat swab of the confirmed COVID-19 patients in Vero CCL81 cells. Viral stocks (3.2 × 10^7^ TCID_50_/mL) were prepared after serial passages in Vero CCL81 cells using DMEM supplemented with 4.5 g/L of glucose and 300 μg/mL L-glutamine, 100 μg/mL penicillin, and 100 μg/mL streptomycin sulfate. Viral stocks used to modulate infection in animals were kept as aliquots at −80 °C.

Madin–Darby canine kidney cells (MDCK) were grown in modified Eagle’s medium (MEM; CellGro, Manassas, VA, USA), supplemented with 10% FBS (fetal bovine serum) and 300 μg/mL L-glutamine, 100 μg/mL penicillin, 100 μg/mL streptomycin sulfate, and 100 μg/mL kanamycin sulfate in a humidified atmosphere of 5% CO_2_. Vero CCL-81 cells (African green monkey kidney epithelial cells) were grown at 37 °C, with 5% CO_2_, in DMEM (PanEko, Moscow, Russia) supplemented with 5% FBS (Invitrogen, Carlsbad, CA, USA); 4.5 g/L of glucose (Sigma-Aldrich, St. Louis, MO, USA); 300 μg/mL of L-glutamine (PanEko, Moscow, Russia); and 100 μg/mL penicillin and 100 μg/mL streptomycin sulfate (PanEko, Moscow, Russia).

Antiviral Preparation. Cesarox Epi further DIM Epi, (LLC Epipharma, Moscow, Russia, PCT/RU2014/000360) was dissolved as a stock 0.2 g/mL in fish oil and diluted to the required doses (100, 60, 50, 20, and 10 mg/kg/day in terms of DIM).

Oseltamivir (LLC IZVARINO PHARMA, Moscow, Russia) was dissolved as a stock 3.6 mg/mL in sterile distilled water and diluted to the required doses (20, 10, and 5 mg/kg/day).

Molnupiravir (LLC OZON, Moscow, Russia) was dissolved as a stock 0.375 g/mL in 1% aqueous starch solution and diluted to the required doses (100 and 200 mg/kg/day).

The stock solution was calculated according to the active substance. For each experiment, freshly made stocks were used.

Animals. Four-to-five-weeks-old female BALB/c mice weighing approximately 14 to 16 g were obtained from the Stezar nursery (Vladimir Region, Russia); 15 animals in each group. Six-weeks-old female Golden Syrian hamsters, 55–60 g, (Mesocricetus auratus) were obtained from the Pushchino nursery (IBCH RAS, Pushchino, Moscow Region, Russia); 10 animals in each group. All animals were quarantined and acclimated for 3 days prior to use. The following conditions were maintained in the facility during the experiment: temperature of 20–24 °C, 12/12 h light cycle, relative humidity of 45–65%, and ventilation without recirculation with an air exchange of 7–12 room volumes per hour. The mice and hamsters were randomized based on their weight and divided into vehicle or treatment groups prior to infection and treatment.

Ethical Considerations. All the animal studies were carried out in according to the “Approval of the Rules of Good Laboratory Practice” (Order of the Ministry of Health of the Russian Federation No. 199n dated 1 April 2016) and the requirements of the European Convention for the Protection of Vertebrate Animals Used by the Institute for Experimental and Other Scientific purposes (Strasbourg, France, 18 March 1986). The research protocol was approved by the Research Ethics Review Committee of the I. I. Mechnikov Research Institute for Vaccines and Sera, Moscow, Russia (approval № 12, 4 October 2023).

Mice model of influenza infection. For influenza infection, mice were randomly divided into virus control and experimental groups (15 animals in each) treated with either a mock injection of fish oil or DIM Epi (10, 20, and 60 mg/kg daily) or Oseltamivir (20 mg/kg daily), or a combination of both, for 5 days. DIM Epi and Oseltamivir were administered orally using lavage in a volume of 100 μL. For acute lethal dose infection, mice were lightly anesthetized and inoculated intranasally with 10 MLD_50_ (mouse lethal dose) of mouse-adapted A/California/04/2009 (H1N1)pdm09 (10^4^ EID_50_) virus in phosphate-buffered saline (PBS). Weight loss and survival were monitored for 15 days after challenge. Mice with weight loss exceeding 30% or mice that were clearly terminal were euthanized in accordance with the guidelines established by the animal ethics committee. At day 4 post-challenge, mouse lungs were collected for viral titration, for which three mice from each experimental and control group were sacrificed; the lungs were removed under sterile conditions and were thoroughly rinsed with 0.01 M sterile PBS, homogenized, and suspended in 1 mL of cold PBS. After separation of the cell debris by centrifugation at 2000× *g* for 10 min, the supernatant was used to determine the viral titer in the MDCK cell culture by the generally accepted method. Virus titers in mouse lungs were calculated as the mean log_10_TCID_50_/mL ± SD.

Hamster model of COVID-19. For SARS-CoV-2, hamsters were randomly assigned into groups (10 animals in each) and treated via oral gavage DIM Epi (50 and 100 mg/kg), Molnupiravir (100 and 200 mg/kg), or double DIM Epi plus Molnupiravir groups, with the same dosage as used in the single treatments. Drugs were administered for 5 days, starting 4 h before infection, DIM Epi once daily, Molnupiravir twice daily (half daily dose per treatment). Drugs with pointed doses were diluted to a volume 100 μL in fish oil (DIM Epi) and 1% starch solution (Molnupiravir). Vehicle control animals received fish oil at the same schedule and in the same volume as treated by DIM Epi groups. For infection, 10^3.5^ TCID_50_ of SARS-CoV-2 virus Dubrovka strain was used. Animals were anesthetized with isoflurane, and 50 μL inoculum was dropped into each nostril by pipette for a total infection volume of 100 μL. Throughout the experiment, the animal weights, temperature, and clinical signs were monitored once daily.

Clinical signs in hamsters were evaluated using scoring system according to A.D. Gruber et al. [21]. Clinical signs in hamsters were estimated daily in accordance with the scoring system based on the Protocol of the clinical study of infectious diseases of small laboratory animals [22] (Table 1).

The weight change was calculated for each hamster as a percentage of its weight on day 0 before virus inoculation. The reported values are average percent changes in weight ± standard errors (SEs). Three and five hamsters from each group were euthanized under anesthesia on day 3 and 7 after infection, respectively, and the lungs were removed under sterile conditions. Virus titer, weight lung index, and pathology were assayed. After washing three times in a solution of PBS, the lungs of each hamster were weighted for the weight lung index (WLI). WLI was calculated using the following formula for each animal separately:WLI=weight of the lung, ganimal weight,g×1000

The average values of the weight lung index ± standard errors (SEs) were determined for each hamster group.

The right lung was placed in a formaldehyde solution for histological analysis. The left lung was homogenized and resuspended in 1 mL of cold sterile PBS. The suspension was cleared of cell debris by centrifugation at 2000× *g* for 10 min, and the supernatant was used to determine the infectious titer of the virus in cell culture.

Virus titration assay. The A/California/04/2009 (H1N1)pdm09 virus and SARS-CoV-2 virus end-point titrations were performed in MDCK and Vero E6 cells, respectively. Cells were inoculated with 10-fold serial dilutions of lung supernatant samples in 100 μL of working media. MDCK and Vero E6 cells were incubated for 3 and 5 days, respectively, and then scored for cytopathogenic effects, and the virus titer was calculated by the Reed–Muench method using an online calculator and expressed as log10 TCID_50_/mL [23].

Lung Histological Examination. Formalin-fixed paraffin-embedded right hamster lungs stained by hematoxylin and eosin were used for histological study. Histological sections were examined using a BX51 light microscope (Olympus, Tokyo, Japan) and photofixed using an Olympus XC10 camera. Pathomorphological changes in the lungs were scored blindly, using a cumulative severity score of 0 to 3 each for an extended range of parameters, as proposed by Gruber A.D. et al. [21]: % of lung area affected, distribution of lesions, necrosis of bronchial ciliated epithelial cells, cellular debris in bronchi, diffuse alveolar damage, necrosis of alveolar cells, cellular debris in alveoli, alveolar hemorrhage, alveolar edema, perivascular/interstitial edema, vasculitis, necrosis and desquamation of vascular endothelial cells, bronchitis, bronchointerstitial pneumonia, interstitial pneumonia, intraalveolar neutrophils and macrophages, hyperplasia of bronchial ciliated epithelial cells, and hyperplasia of type II alveolar cells. The maximum possible number of scores was 60 [21].

Lung Visual Examination. Visual assessment of lung lesions was performed by two independent researchers, and both results were represented in the data. The percentage of damage was determined by the total number of foci of hemorrhagic pneumonia in the lungs. A 100% lesion was considered to be when the foci of hemorrhagic pneumonia occupied the entire area of the lobes of the lungs, 50% when the sum of the lesion areas occupied half of the total area of the lung lobes; further, the percentages indicated correspond to the percentages of the foci of hemorrhagic pneumonia on the section.

Statistical Analysis. The statistical analysis was carried out using the R-Studio (version 2024.04.2) and the “drc” package. The data were presented as an average value of ±standard deviation (SD). The normality of the data distribution was assessed using the Shapiro–Wilk test. The Kruskal–Wallis criterion with Bonferroni correction was used to compare the statistical significance of the differences between the experimental and control groups. The differences were considered significant if *p* < 0.05. The three-parameter log-logistic (LL.3) function was used. Estimating effective doses, namely 50% cytotoxic concentration (CC_50_) and 50% effective concentration (EC_50_), was undertaken with asymptotics-based confidence intervals. A *p*-value of <0.05 was considered statically significant. The weight change was calculated for each hamster as a percentage of its weight on day 0 before virus inoculation. The reported values are average percent changes in weight + standard errors (SEs). For a score assessment of the condition of animals and affected lungs, visual and microscopic examination methods were used; the scores obtained in the therapy groups were statistically compared with the viral control group using the Kruskal–Wallis criterion with Bonferroni correction. The difference was considered statistically significant if *p* < 0.05.

## 3. Results

Efficacy of Diindolylmethane monotherapy against influenza pneumonia in mice. First, we investigated whether Diindolylmethane (DIM Epi) therapy could protect mice from lethal influenza virus challenge. In the preliminary experiments, DIM Epi was administered to intact mice in doses up to 60 mg/kg/day, according to the schemes suggested for further treatment, without occurrence of acute toxic effects. The antiviral effects of DIM Epi in mice were examined using the experimental scheme shown in Figure 2A. After infection with 10 MLD_50_ of the A/California/04/2009 (H1N1)pdm09 virus, which corresponds to 10^4.5^ TCID_50_/0.1 mL, mice were treated with DIM Epi (10, 20 and 60 mg/kg daily) once a day for 5 days. The first injection of drugs was 4 h before infection. In the control group of non-treated infected mice, all mice died by day 12 after infection; MSD in this group was 7.0 days (Figure 2B). Infection with the virus also caused significant weight loss in mice; on the 10th day after infection, it was about 25% (Figure 2C). The viral titer of the lungs of animals in this control group on day 4 after infection with influenza A/California/04/09 (H1N1) virus was high, exceeding >7.0 lg TCID_50_/0.1 mL, which caused severe viral pneumonia and the death of the animals (Figure 2D). The treatment by DIM Epi in all the studied doses was effective and depended on doses increasing with their increase. The administration of DIM Epi at doses of 10 and 20 mg/kg/day statistically significantly reduced mortality (by 42%) and weight loss, as well as increased MSD by 1.7 times. The most effective was treatment with the highest dose of DIM Epi (60 mg/kg/day), which protected 83% of the mice from death and weight loss (Figure 2B,C). The MSD was 2-times higher compared to the viral control group. This treatment was comparable to treatment with Oseltamivir at dose 20 mg/kg daily, which also increased the survival rate to 75% and MSD by 2 times and protected against weight loss (Figure 2A,B). In addition, treatment with Oseltamivir more than two-times inhibited virus replication in mouse lungs compared to untreated mice (3.42 ± 0.38 vs. >7.0 lg TCID_50_/0.1 mL) (Figure 2D). However, although the titer of the virus in the lungs during treatment with all doses of DIM Epi was slightly lower than in the viral control group, a statistically significant difference in virus titers was observed only in the group with DIM Epi therapy at a dose of 60 mg/kg/day (5.25 vs. >7.0 lg TCID_50_/0.1 mL).

Antiviral efficacy of combination therapy by DIM Epi and Oseltamivir. To reveal the effect of combination of Diindolylmethane (DIM Epi) with direct antiviral we decrease the doses of Oseltamivir to 5 and 10 mg/kg/day (Figure 3A). Monotherapy with Oseltamivir at doses of 5 and 10 mg/kg/day protected 40–50% of the animals from death, reduced weight loss, and increased MSD by 1.7–1.9 times (Figure 3B,C). In addition, the treatment with Oseltamivir statistically significantly (more than 2 lgTCID_50_) inhibited the replication of the virus in the lungs of the animals (Figure 3D). Combination therapy resulted in improved efficacy in the infected mice. The same Oseltamivir dosages when coupled with 10 and 20 mg/kg/day DIM Epi reduced body weight loss in mice (Figure 3C) and resulted in respective 77% and 83% survival rates (Figure 3B). The mean days until death were also lengthened with this dosage of Oseltamivir administered in combination therapy. Interestingly, combinations of Oseltamivir in doses 5 and 10 mg/kg/day with increased concentrations of DIM Epi 60 mg/kg/day produced even better therapeutic efficiencies and prevented death and body weight loss of the mice and also almost completely suppressed the replication of the virus in the lungs of the animals. The efficacy of the studied combinations was comparable and even exceeded the efficacy of Oseltamivir at high dose (20 mg/kg/day), which was used as a control antiviral in monotherapy experiments. No in vivo toxicity was observed in mice for all the studied combinations. Considered together, these results indicate that the dearth in the monotherapeutic efficacy of drugs can be compensated for by using combination protocols, and an adequate combination DIM Epi with Oseltamivir may enhance the survival rates of mice against influenza virus (Figure 3B–D).

Efficacy of DIM Epi mono- and combination therapy against SARS-CoV-2 virus infection in hamsters. The inhibitory effects of DIM Epi against influenza pneumonia observed in mice prompted us to evaluate the efficacy against COVID-19 infection induced by SARS-CoV-2. We assessed the therapeutic efficacy of DIM Epi in hamsters infected with the Dubrovka (Wuhan-like) variant. Hamsters inoculated intranasally with 10^3.5^ TCID_50_ were treated via oral gavage once daily for 5 consecutive days, beginning 4 h before viral infection with 50 mg/kg/day or 100 mg/kg/day of DIM Epi (Figure 3A). These dosages were selected on the basis of previous studies in mice infected with influenza viruses. Molnupiravir taken as a control antiviral was administered via oral gavage once daily for 5 consecutive days, beginning 4 h before viral infection at doses 100 or 200 mg/kg/day.

On the 3rd day after infection, pronounced clinical signs of respiratory disease began to appear in the virus control group. The animals showed frequent prolonged sneezing with abundant mucosal exudate. The nasal mucosa of the hamsters was reddened during clinical examination, and mild edema was noted (Table 2). The animals were actively engaged in grooming, especially strongly wiping the nasal mirror. Physiological activity in the viral control group was moderately slow—the animals took food and water but reacted reluctantly to stimuli (Table 2). From the 4th day of the study, an increase in temperature above the physiological norm was observed in this group of animals. The severity of clinical signs of the disease in the viral control group is estimated in total at 12 points (Table 2).

Animals with monotherapy with DIM Epi 50 and 100 mg/kg/day and Molnupiravir 100 and 200 mg/kg/day had similar clinical manifestations of the disease, which were estimated at 2–3 points. In hamsters, there was a slight hyperemia of the nasal mucous membranes and a slight increase in the secretion of the nasal glands, manifested by small, rare drops of transparent exudate around the nasal mirror. Animal activity and interest in water and food were normal, and no acts of sneezing or coughing were observed throughout the observation period. The use of a combination of drugs at the doses (Molnupiravir 100 mg/kg/day + DIM Epi 50 mg/kg/day; Molnupiravir 100 mg/kg/day + DIM Epi 100 mg/kg/day; Molnupiravir 200 mg/kg/day + DIM Epi 50 mg/kg/day; Molnupiravir 200 mg/kg/day + DIM Epi 50 mg/kg/day) did not lead to a change in the clinical signs of infection compared with monotherapy with these drugs at the same doses. The nasal mirror of the hamsters in these groups was slightly hyperemic and had a bright pink hue; drops of transparent exudate accumulated around the mirror; the animals actively wiped it during grooming, sneezed 3–5 times within 10 min. At the same time, the activity of animals, interest in water and food, response to stimuli, and interest in changing the situation corresponded to the physiological norm and did not differ from the group of intact animals. Coughing or wheezing was also not observed in these groups. In general, the clinical signs of infection in these groups were estimated at 2–4 points (Table 2).

The severity of respiratory infection in the virus control group decreased by day 7. Sneezing was still observed in the animals, there were large drops of transparent exudate in the nasal area, and the nasal mucosa and nasal mirror were bright red, which in general allowed us to assess the condition of the animals at 9 points. At the same time, in all the treated groups, the severity of clinical signs decreased to 1–2 points, which indicated the resolving stage of the infectious process and recovery. By day 7, there was no difference between the groups treated with DIM Epi and Molnupiravir and groups with monotherapy and combined therapy.

Body weight and body temperature were monitored daily as the clinical symptom markers. Monitoring of hamster weight daily showed that the placebo-treated infected hamsters experienced severe weight loss (Figure 4B), losing an average of 17.5% by day 7. Treatment with drugs in all tested doses and combinations revealed a statistically significant decrease in weight loss compared with the viral control group (*p* < 0.05). Thermometry of the hamsters was performed rectally, in the morning, before the procedures, in order to exclude falsely inflated body temperature indicators associated with the stress factor. In the viral control group, on day 3 after infection, by the time of the manifestation of general clinical signs of an infectious disease, the body temperature of the hamsters was 38.8 °C, which is 0.3 °C higher than normal. On day 4 after infection, the body temperature of animals from the viral control group reached a peak and averaged 39.0 °C (exceeding the physiological norm by 0.5 °C), which is clinically significant for rodents and indicates a high intensity of the infectious process. Animals in all therapy groups showed an increase in temperature from day 3 to day 5 of the experiment, with the peak increase occurring on day 3–4 in hamsters, and then the body temperature decreased. During the period of observation, the body temperature of the hamsters in the treatment groups was within the physiological norm (37.5–38.5). The animals in the therapy groups showed an increase in temperature from day 3 to day 5 of the experiment, with the peak on days 3–4; the temperature increase was bordering on the maximum permissible physiological norm but did not exceed it. The body temperature in all the treated groups decreased, beginning from day 5. The difference between the treated groups and the viral control group on day 4 after infection was significant (*p* = 0.0000529), and no difference between body temperatures in the treated groups was revealed (Figure 4C).

Antiviral effects of mono- and combination therapy on the lungs of infected hamsters. Severe pathogenesis in the lower respiratory tract is one of the most important features of SARS-CoV-2- virus infection in humans. Therefore, we assessed whether DIM Epi therapy alone or in combination with Molnupiravir could counteract COVID-19 infection in the lungs of hamsters. For this estimation, the following criteria were used: viral titer and visual and histological lung examination.

Virus titration. On day 3 after infection, in the lungs of the placebo-treated hamsters, the SARS-CoV-2 virus was able to replicate at 8.05 lg TCID_50_/mL (Figure 5A). In the hamsters with DIM Epi monotherapy at a dose of 50 and 100 mg/kg/day, and with Molnupiravir monotherapy at a dose of 100 mg/kg/day, the titer of the virus was slightly lower but did not differ significantly from the titer in the viral control group (Figure 5). When treated with Molnupiravir at 200 mg/kg/day, as well as in all groups treated with drug combinations, viral replication was alleviated (5.39–5.94 lgTCID_50_/0.1 mL vs. 8.05 lgTCID_50_/0.1 mL, Figure 5). On day 7 after infection, when the lung condition in all groups was visually assessed as being in the resolution stage (see below), the titer of the virus in the lungs was reduced both in the viral control group and in all groups of animals treated separately with DIM Epi or Molnupiravir, and all their combinations (2.5 lgTCID_50_/0.1 mL) (Figure 5A).

For estimation of the severity of pneumonia, we used additional criteria that quantitatively reflects the degree of the pulmonary edema and the inflammatory process, weight lung index (WLI).

Determination of WLI in all groups on the 3rd day after infection showed that WLI was significantly lower in groups treated by DIM Epi in all studied doses and combinations of DIM Epi with Molnupiravir compared with the viral control group (*p* ≤ 0.05) (Figure 5B). At the same time, there was no significant difference between WLI along all the treated groups.

Pulmonary Pathological Changes in Hamsters by Visual Examination. Following the harvest of lung tissues from hamsters, examinations were performed in order to evaluate pathological changes caused by infection with SARS-CoV-2 and to determine whether mono- or combined treatment with DIM Epi and Molnupiravir can inhibit pathological changes. In the viral control group, a pathoanatomic macroscopic examination of the lungs revealed widespread extensive foci of hemorrhagic drain pneumonia, occupying 60–70% of the total lung volume (Table 3). Lung tissues were destroyed in some places and had a softened structure. Visual pathoanatomical macro-examination of the lungs showed that all studied doses and combinations of the studied drugs had a therapeutic effect. With monotherapy, the lungs were least affected when treated with DIM Epi at doses of 50 and 100 mg/kg/day. Foci of hemorrhagic drain pneumonia occupied up to 10% of the total lung volume area (Table 3). Moreover, in some animals, the foci of pneumonia were not externally visualized at all. The lungs were smooth, elastic, and of normal consistency and color. During monotherapy with Molnupiravir at a dose of 100 mg/kg/day, inflammatory processes were observed in lung tissues as hemorrhagic drain pneumonia, occupying 20–25% of the total lung volume. Tissue destruction and lung softening were not found in the group. An increase in the dose of Molnupiravir to 200 mg/kg/day led to an improvement in the condition of the lungs, reducing the volume of hemorrhagic drain pneumonia to 15–20%. Combination of DIM Epi at doses of 50 mg/kg/day and 100 mg/kg/day with Molnupiravir at doses of 100 mg/kg/day and 200 mg/kg/day did not improve the lung condition compared with DIM Epi monotherapy at these doses; pathological signs in the lungs in these groups were similar (Table 3).

On day 7 of viral pneumonia, the lungs of experimental hamsters in each group were visualized at the stage of disease resolution. There were single fibrin nodules at the sites of focal and diffuse pneumonia. In places, a point accumulation of erythrocytes was observed in the marginal zones, occupying no more than 0.5 mm, which indicated the completion of the inflammatory process in these areas of the lungs. Visually, there were no obvious differences in the severity of inflammatory manifestations at this stage between the viral control group and the groups using mono- and combination therapy of both drugs in all studied doses.

Histological lung examination. The data obtained were confirmed by histological examination of the right lung (Figure 6). Since a visual examination of the lungs taken on day 7 after infection showed a pneumonia resolution and did not reveal difference between the lungs of treated and untreated animals, we performed a histological examination of the lungs taken only on day 3 after infection. At 3 dpi, the lung sections of the untreated SARS-CoV-2-infected hamsters showed marked alterative and inflammatory changes, which morphologically corresponded to severe bronchointerstitial pneumonia, the foci of which were located in all lobes of the right lung and occupied up to 80% of the total area of the histological section of the organ (Figure 6a), compared to the lungs of uninfected animals (Figure 6j). Pronounced inflammatory changes were observed in the walls of the bronchi, bronchioles, and their accompanying vessels. The respiratory tract in the foci of pneumonia looked airless; the alveolar lumen was not detected here; the interalveolar septa were destroyed due to the death of alveolocytes and a pronounced lymphoid-histiocytic infiltrate with a small admixture of neutrophils and eosinophils. In more recent areas of pneumonia, the alveolar lumens were filled with inflammatory exudate and cellular debris. The interalveolar septa looked thickened due to capillary congestion, interstitial edema, and lymphoid-histiocytic infiltration.

The histological examination of lung tissue of hamsters treated with DIM Epi at doses of 50 and 100 mg/kg/day and Molnupiravir at doses of 100 and 200 mg/kg/day alone at 3 dpi showed a decrease in the severity of the alterative inflammatory changes and the area of airless foci of pneumonia to 20–30%, versus 80% in the viral control group (Figure 6b–e). In the respiratory region of all lobes of the lung, mainly small foci of bronchointerstitial pneumonia were observed, located along the bronchial tree. Alterative inflammatory changes in the walls of the bronchi, bronchioles, and blood vessels were expressed moderately, compared with those in the viral control group.

A pronounced effect was also noted in the groups treated with combinations of drugs (DIM Epi at a dose of 50 mg/kg/day + Molnupiravir 100 mg/kg/day; DIM Epi at a dose of 100 mg/kg/day + Molnupiravir 200 mg/kg/day) In these groups, the area of pneumonia foci did not exceed 20% of the total area of the histological section of the organ. In the groups treated with a combination of the drugs DIM Epi and Molnupiravir, microscopy revealed mainly small peribronchial foci of bronchointerstitial pneumonia located along the large bronchi in histological sections. Moderate inflammatory changes were observed in the wall of these bronchi; the epithelial lining was preserved almost throughout; dystrophic changes and necrosis of epitheliocytes were focal. There were no inflammatory changes in the vessel walls (Figure 6f–i).

Based on histological examination, the pathological changes in the lungs of the animals were given a score according to Gruber A.D. et al. [21] (Figure 7). In the untreated animals challenged with SARS-CoV-2 virus, severity score on day 3 after infection was 48.3. At the same time, the infected animals treated with DIM Epi at doses 50 and 100 mg/kg/day had significantly lower lungs histopathology scores, 29.9 (*p* < 0.05) and 38.1 (*p* < 0.05), respectively, compared to hamsters from the virus control group (Figure 7). Treatment with Molnupiravir at doses 100 and 200 mg/kg/day reduced the lungs histopathology score by 1.9–1.8 times, 25.7 score and 27.4 score (*p* < 0.01), respectively, compared to untreated hamsters (Figure 7).

## 4. Discussion

Influenza and SARS-CoV2 infections are often accompanied by aggressive pro-inflammatory responses coupled with inadequate control of the anti-inflammatory mechanisms, leading to a combination of events known as the “cytokine storm” [24]. 3,3′-Diindolylmethane (DIM Epi) is a naturally occurring substance with unique multi-target activity. DIM Epi specifically acts (including at the epigenetic level) on a wide range of pathogenetic mechanisms induced by the host as a result of viral invasion. DIM Epi suppresses an abnormal inflammatory reaction (cytokine storm), improves the immune and antioxidant response, blocks the development of hypercoagulation and the spread of microthrombosis, and has antibacterial activity. In this study, we evaluated the therapeutic effect of DIM Epi in vivo in a murine model of lethal influenza A (H1N1) infection and in a hamster model of SARS-CoV-2 infection induced by the Dubrovka (Wuhan-like) strain. The therapeutic effect of oral DIM Epi administration following A (H3N2) challenge in a lethal infection murine model was shown earlier [20]. In a previous report, 5 days of dosing of DIM Epi protected mice infected with A/Aichi/2/68 (H3N2) from death. Comparable results were obtained in this study. In our model of lethal influenza pneumonia induced by influenza A/California/04/2009 (H1N1)pdm09 virus, we showed that 5 days of treatment with DIM Epi at 10, 20, and 60 mg/kg/day delayed the time to death, prevented body weight loss, and resulted in significant improvements in survival. Similarly, in the hamster model of SARS-CoV-2 infection, treatment with DIM Epi (50 and 100 mg/kg/day) was associated with clinical benefits. Animals in groups treated by DIM Epi experienced reduced weight loss and body temperature and mitigated lung pathology compared with control animals.

We also measured the virus titer to determine whether these clinical symptoms improvements in the DIM Epi-treated groups was due to the difference in viral titers between groups. In both models of infections, measurement of the viral titer in the animal lung tissues did not show a significantly decreased viral titer in the groups treated with DIM Epi compared to that in the viral infection groups. However, in the case of influenza virus, treatment with DIM Epi protected mice from death. The H&E staining of hamster lung tissue also revealed an improvement in lung lesions and as well as an alleviation of pathological sequelae, indicating a therapeutic action against SARS-CoV-2. These outcomes are relevant to note because an important desired effect of antiviral treatment in human COVID-19 and in influenza would be to prevent or reduce the progression of virus-induced pneumonia. Collectively, these results support our hypothesis that DIM Epi may enhance outcomes in respiratory infections by modulating inflammation rather than directly suppressing viral replication.

There is arguably a consensus among virologists that combining two compounds with different viral or cellular targets could result in better virological and physiological responses than those of antiviral monotherapy, for which we decided to explore this hypothesis in the case of DIM Epi. In this study, we tested a combination of DIM Epi with direct-acting antivirals—Oseltamivir against influenza infection and Molnupiravir against SARS-CoV-2 infection. We did not reveal an antagonistic interaction at different doses of the studied combinations. In the mouse model of influenza infection, we demonstrated that combination treatment can have a greater antiviral effect than monotherapy in vivo (*p* ≤ 0.05). DIM Epi and Oseltamivir were more effective against infection caused in mice by mouse-adapted A/California/04/2009 (H1N1) influenza virus when given in combination than given singly. On the contrary, in the hamster model of SARS-CoV-2 infection, the combination of DIM Epi with Molnupiravir did not significantly improve outcomes compared to either agent alone. The discrepancy in effects in the combination treatment may be attributed to differences in the animal models of infection. The Syrian golden hamster is a suitable animal model for the evaluation of effective treatments for SARS-CoV-2 infection in that efficacy is mainly estimated by histopathological features observed in lung tissues of infected animals that are similar to COVID-19-associated pneumonia in humans. The limitation of this estimation is that a strong quantitative evaluation of the histopathology features observed in the lungs is not included. It is very different from peracute and uniformly lethal mouse models of influenza infection in that efficacy is estimated by protection from death. The study of the efficacy of DIM Epi in a lethal model of SARS-CoV-2 infection using transgenic mice which express human ACE2 would be very desirable.

In addition, there are some limitations in our study that may explain the lack of effect of combination in the SARS-CoV-2 hamsters. Measurement of the temperature (fever, one of the estimated clinical signs) in hamsters infected with SARS-CoV-2 is not possible in our laboratory, but telemetry probably would be a more accurate measurement of temperature. Using it could reveal significant differences with monotherapy and combination therapy. Another important factor is the use of a small number of animals in each group, and all reported outcomes must be carefully considered. Due to the small number of samples, discrepancies in statistical analyses may have arisen because of a lack of power and high variability within groups. One other limitation of our study is the use of Molnupiravir as the reference antiviral in the SARS-CoV-2 model, which may have lower clinical efficacy compared to nirmatrelvir/ritonavir.

An important mechanism for the development of viral infection is the ability of the virus to induce epigenetic transformations in the affected organ. It is obvious that the severity of the infection is mainly determined by the capacity of viruses to cause paralysis of the innate immune system and induce hyper-inflammatory reactions [25]. The phenomenon of epigenetic reprogramming has been convincingly demonstrated in the case of SARS-CoV-2 [26]. It was found that the virus is capable of causing changes in the three-dimensional structure of the genome/epigenome, which led to suppression of transcription of interferon response genes, paralyzing the response of innate immunity to viral infection. At the same time, the expression of genes of pro-inflammatory cytokines induced during severe COVID-19 was sharply activated.

DIM, the active compound in the tested CesaroxEpi formulation, demonstrated multi-target activity across numerous pathogenic pathways associated with viral infections. Extensive in vitro, in vivo, and in silico studies have shown that DIM exhibits antiviral, anti-inflammatory, antioxidant, antifibrotic, and antithrombotic properties [17,18,26,27,28,29,30,31,32,33,34,35,36,37,38,39,40,41,42,43,44,45,46,47,48,49,50,51,52,53]. It suppresses replication of respiratory viruses, including influenza A virus, and attenuates cytokine storm by downregulating pro-inflammatory mediators such as IL-1β, IL-6, TNF-α, COX-2, and key transcription factors (e.g., NF-κB, STAT3) [36,37,38,39,40,41,42]. Mechanistically, DIM modulates multiple signaling pathways including EGFR, PI3K/Akt, Raf/MEK/ERK, Wnt/β-catenin, HIF-1α, ICAM-1, and JAK/STAT [17,27,28,29,30,31,32,33,34,35]. It also inhibits immune checkpoints PD-1 and PD-L1 [31,36,37,41,43,44] and regulates T cell polarization by balancing Th17/Treg cell populations [44,45,46]. Moreover, DIM activates antioxidant defense mechanisms via Nrf2 and BRCA1-mediated signaling [41,47,48,49,50,51,52]. In the context of lung protection, DIM exhibits antifibrotic effects and modulates post-infectious tissue remodeling [18,37]. It also demonstrates antithrombotic activity by inhibiting platelet aggregation induced by ADP, collagen, thrombin, and arachidonic acid, reducing thrombus size, and delaying vascular occlusion time without thrombolytic action [53]. These effects are associated with reductions in ROS, H_2_O_2_, COX-1, TXB_2_, and PGE_2_ levels and increases in cAMP [53]. In addition, hypermethylation of genes in lung tissue was found in patients who died from viral infection, which, according to the authors, was the cause of death [54]. Epigenetic activity has been established for DIM, which manifests itself both in the ability to regulate the activity of histone deacetylases affecting the chromatin conformation of an infected cell and an inhibitory ability against DNA methyltransferases [55,56]. Another unique epigenetic mechanism is the synthesis of microRNAs induced by viral agents involved both in the infectious process and in the development of post-infectious complications [57]. In particular, microRNA-21 plays an important role in the development of hyper-inflammatory reactions and fibrous changes in the lungs during inflammation [58,59]. DIM is also a unique molecule with direct inhibitory activity against microRNA-21 [60], and this may be a possible mechanism through which the therapeutic activity of DIM is realized. Today, epigenetic therapy is considered as a very promising direction in the treatment of viral infections [61,62].

In conclusion, our data demonstrate the therapeutical potential of DIM Epi in preventing pneumonia associated with influenza and SARS-CoV2 infections. The results obtained provide support for further investigation of DIM Epi as a treatment for viral pneumonia. Furthermore, these outcomes and the mechanism of action of DIM Epi suggest its broad-spectrum antiviral activity against viruses causing viral pneumonia and the role of DIM Epi as a potential tool against a wide range of infectious threats.

## Figures and Tables

**Figure 1 viruses-17-00964-f001:**
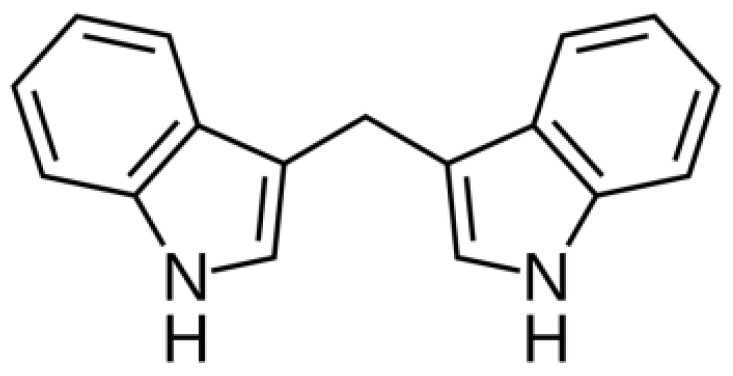
3,3′-Diindolylmethane (IUPAC name—3,3′-Methylenedi (1H-indole); other name—DIM).

**Figure 2 viruses-17-00964-f002:**
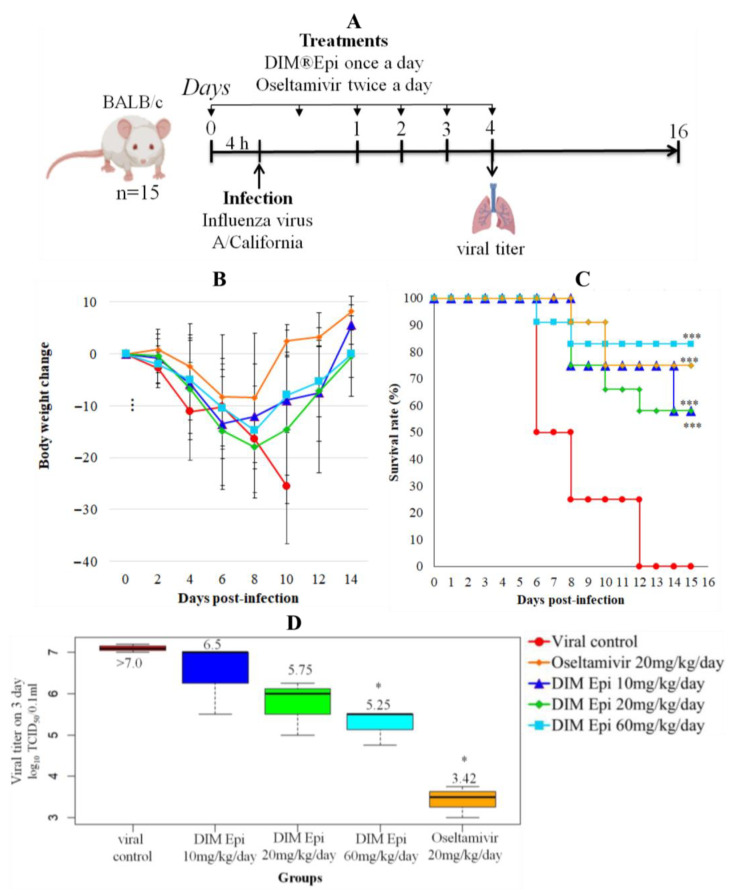
In vivo efficacy of DIM Epi monotherapy as a treatment for influenza pneumonia (n = 15). (**A**) Study design. Four-to-five-weeks-old female BALB/c mice were intranasally infected with 10^4.5^ TCID_50_ influenza A/California/04/2009 (H1N1)pdm09 virus in the presence or absence of DIM Epi or Oseltamivir. Treatment with DIM Epi was performed orally once a day, Oseltamivir twice a day. At 4 dpi, viral titers in the lungs were determined. Survival rate (**B**) and body weight change (**C**) were monitored daily after the virus infection. Lung virus titers were measured at day 4 after infection (**D**). Asterisks denote significant difference from viral control by nonparametric Kruskal–Wallis test with Bonferroni correction, * *p* ≤ 0.05; *** *p* ≤ 0.0001.

**Figure 3 viruses-17-00964-f003:**
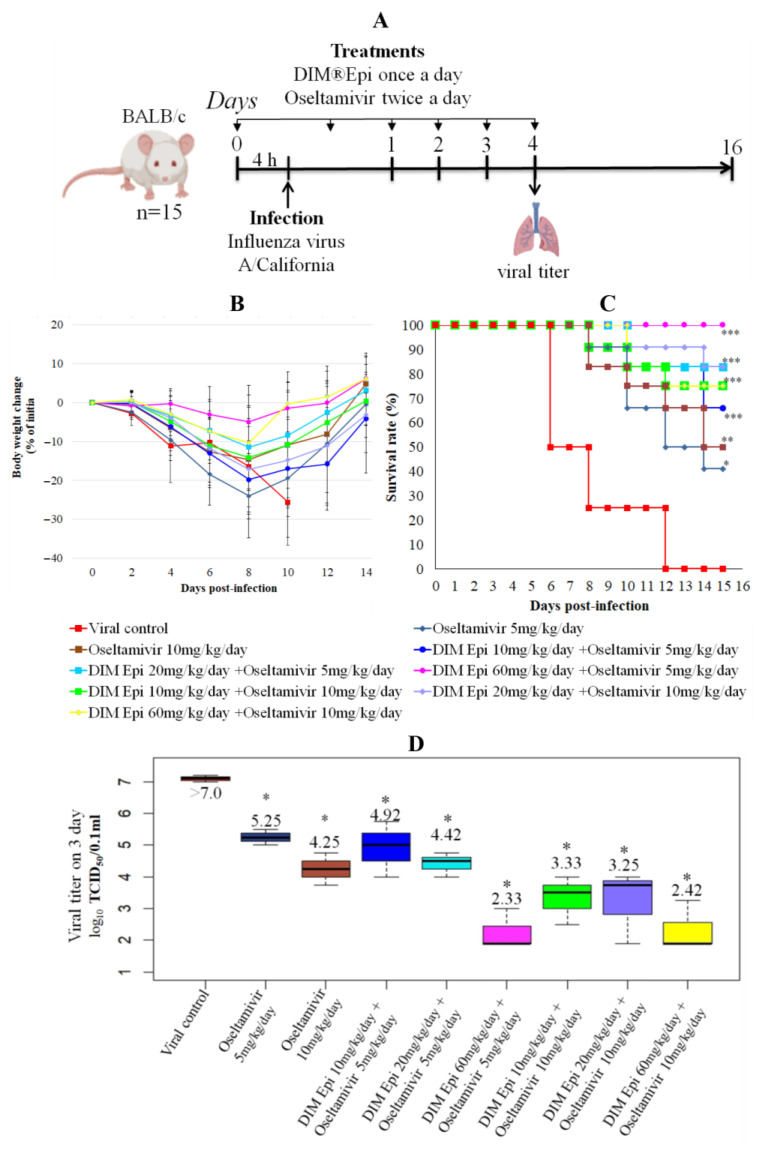
In vivo efficacy of combination therapy by DIM Epi and Oseltamivir as a treatment for influenza pneumonia (n = 15). (**A**) Study design. Four-to-five-weeks-old female BALB/c mice were intranasally infected with 10^4.5^ TCID_50_ influenza A/California/04/2009 (H1N1)pdm09 virus in the presence or absence of DIM Epi and Oseltamivir. Treatment with DIM Epi was performed orally once a day, Oseltamivir twice a day. At 4 dpi, viral titers were determined in the lungs. Survival rate (**B**) and body weight change (**C**) were monitored daily after the virus infection. Lung virus titers were measured at day 4 after infection (**D**). Asterisks denote significant difference from viral control by nonparametric Kruskal–Wallis test with Bonferroni correction, * *p* ≤ 0.05; ** *p* ≤ 0.001; *** *p* ≤ 0.0001.

**Figure 4 viruses-17-00964-f004:**
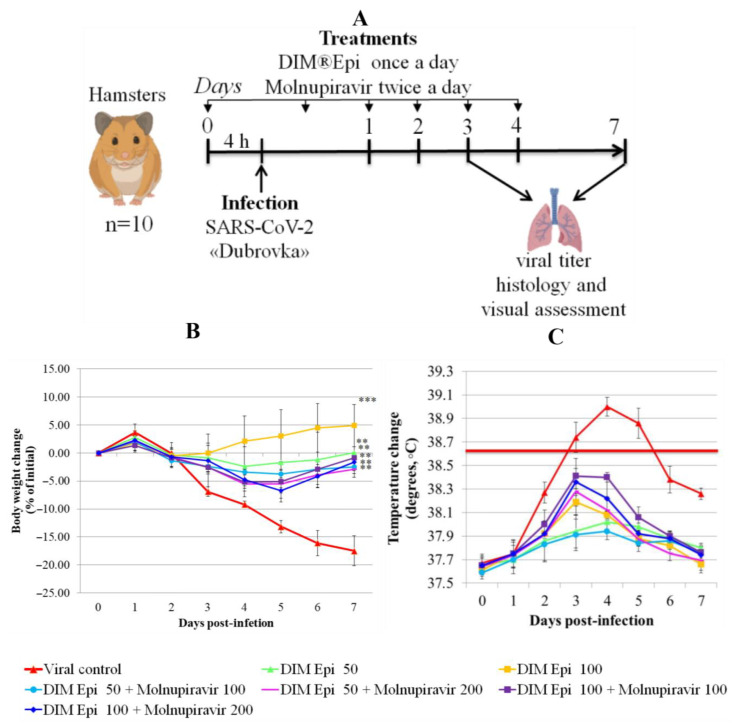
In vivo efficacy of mono- and combination therapy by DIM Epi and Molnupiravir on the lungs as a treatment for SARS-CoV-2 infection (n = 10). (**A**) Study design. Six-weeks-old female Golden Syrian hamsters were intranasally infected with 10^3.5^ TCID_50_ SARS-CoV-2. Treatment with DIM Epi was performed orally once a day, Molnupiravir twice a day. At 3 dpi and 7 dpi, viral titer, histology and pathomorphological changes were determined in the lungs. (**B**) Weight change of hamsters infected with SARS-CoV-2 in percentage, normalized to the body weight at the time of infection. Asterisks denote significant difference from viral control by nonparametric Kruskal–Wallis test with Bonferroni correction, ** *p* ≤ 0.001; *** *p* ≤ 0.0001. (**C**) Body temperature change of hamsters infected with SARS-CoV-2 in degrees. A red dotted line marks the upper limit of the normal body temperature of hamsters.

**Figure 5 viruses-17-00964-f005:**
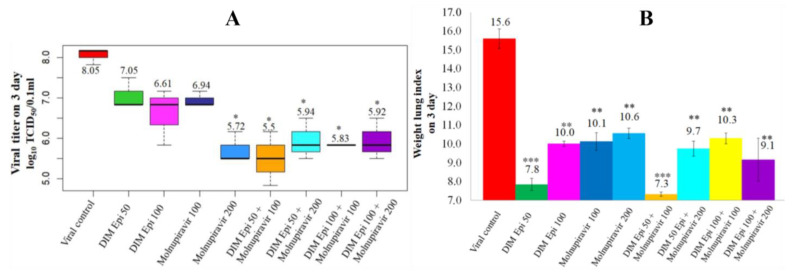
In vivo efficacy of mono- and combination therapy by DIM Epi and Molnupiravir on the lungs as a treatment for SARS-CoV-2 infection (n = 10). Viral titer (**A**) and mass coefficient of the lung (**B**) on day 3 post-infection. Asterisks denote significant difference from viral control by nonparametric Kruskal–Wallis test with Bonferroni correction, * *p* ≤ 0.05; ** *p* ≤ 0.001; *** *p* ≤ 0.0001.

**Figure 6 viruses-17-00964-f006:**
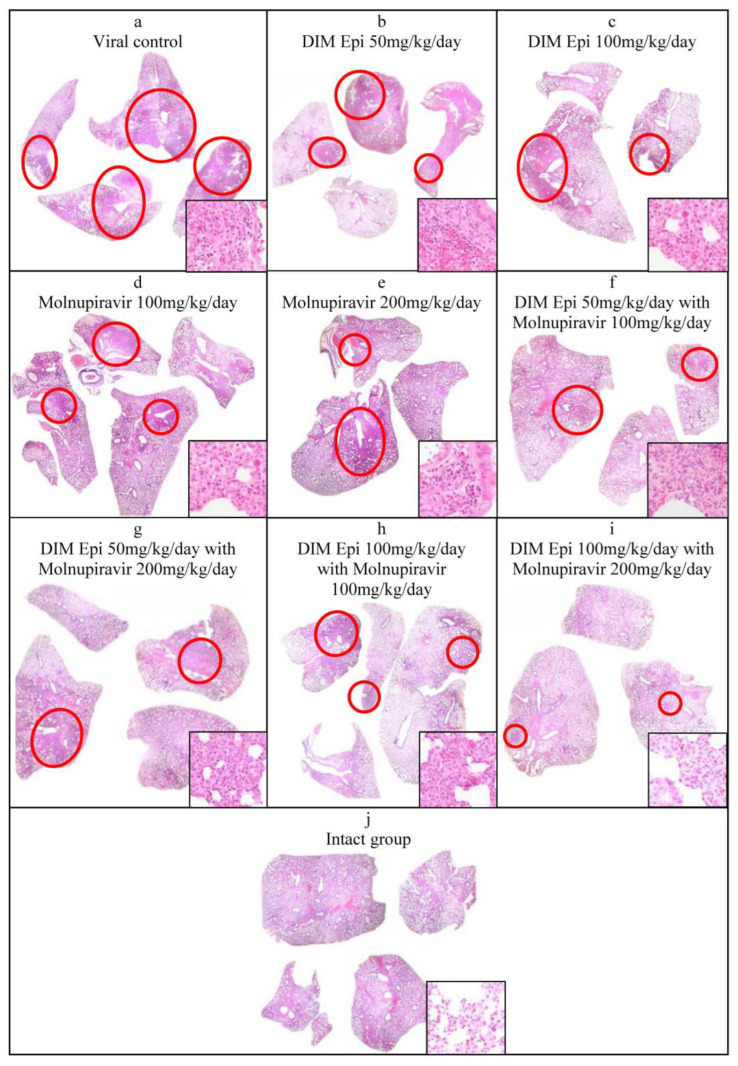
Morphological changes of hamsters’ lungs at 3 days post-infection with SARS-CoV-2 and combined or monotherapy with DIM Epi, n = 3 Circle outlines highlight the foci of bronchopneumonia (20×); additional inserts show enlarged foci of pneumonia in infected groups (**a**–**i**) and lungs of intact animals (**j**) (200×). Cross sections stained with hematoxylin and eosin X20 and X200: (**a**)—viral control; (**b**)—DIM 50 mg/kg/day; (**c**)—DIM 100 mg/kg/day; (**d**)—Molnupiravir 100 mg/kg/day; (**e**)—Molnupiravir 200 mg/kg/day; (**f**)—DIM 50 mg/kg/day + Molnupiravir 100 mg/kg/day; (**g**)—DIM 50 mg/kg/day + Molnupiravir 200 mg/kg/day; (**h**)—DIM 100 mg/kg/day + Molnupiravir 100 mg/kg/day; (**i**)—DIM 100 mg/kg/day + Molnupiravir 200 mg/kg/day; (**j**)—intact group. (**b**–**i**)—Reducing severity and prevalence of the inflammatory process.

**Figure 7 viruses-17-00964-f007:**
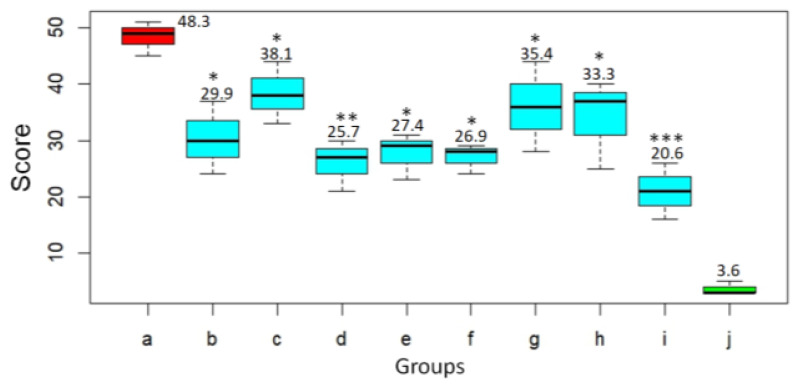
Histopathology score at 3 dpi for hamster lungs challenged with SARS-CoV-2 virus and treated with DIM Epi and Molnupiravir (the maximum total score is 60) (n = 10): a—viral control; b—DIM Epi 50 mg/kg/day; c—DIM Epi 100 mg/kg/day; d—Molnupiravir 100 mg/kg/day; e—Molnupiravir 200 mg/kg/day; f—DIM Epi 50 mg/kg/day with Molnupiravir 100 mg/kg/day; g—DIM Epi 50 mg/kg/day with Molnupiravir 200 mg/kg/day; h—DIM Epi 100 mg/kg/day with Molnupiravir 100 mg/kg/day; i—DIM Epi 100 mg/kg/day with Molnupiravir 200 mg/kg/day; j—intact group. Asterisks denote significant difference from viral control by nonparametric Kruskal–Wallis test with Bonferroni correction, * *p* < 0.05; ** *p* < 0.01; *** *p* < 0.001.

**Table 1 viruses-17-00964-t001:** Scoring system for clinical signs in hamsters infected with SARS-CoV-2.

Characteristics	Scores
Sneezing	Normal—0. Single, repeated less than 1 time in 10 min—1. Single, repeated 1 or more often every 10 min—2. Multiple attacks following each other, repeated once every 10 min—3. Multiple attacks following each other, repeated more than once every 10 min—4.
Cough	Normal—0. Single, repeated less than 1 time in 10 min—1. Single, repeated 1 or more often every 10 min—2. Multiple attacks following each other, repeated once every 10 min—3. Multiple attacks following each other, repeated more than once every 10 min—4.
Secretion of the nasal glands	Normal—0. Mild—1. Moderate—2. Significant—3.
Hyperemia of the mucous membranes	Normal—0. Mild—1. Moderate—2. Significant—3.
Animal activity	Normal—0. Weakening (animals show moderate interest in water, food, respond to stimuli slowly, within a few seconds, adapt quickly to a change of environment)—1. Moderately slow (animals show moderate interest in water, food, respond to stimuli with reluctance, the first seconds do not adapt well to a change of environment)—2. Inhibited (animals are poorly interested in food and water, respond poorly to stimuli, do not show interest in the environment when the situation changes)—3.

**Table 2 viruses-17-00964-t002:** A score assessment of the clinical condition of hamsters infected with SARS-CoV-2 (n = 10).

	Day 3									Day 7								
	Viral control	Monotherapy				Combination				Viral control	Monotherapy				Combination			
		D50	D100	M100	M200	D50 + M100	D50 + M200	D100 + M100	D100+ M200		D50	D100	M100	M200	D50 + M100	D50 + M200	D100 + M100	D100 + M200
Sneezing	4	0	0	0	0	0	2	2	2	2	0	0	0	0	0	0	0	0
Cough	2	0	0	0	0	0	0	0	0	0	0	0	0	0	0	0	0	0
Secretion of the nasal glands	2	1	1	1	2	1	1	1	1	2	0	0	0	0	0	1	0	0
Hyperemia of the mucous membranes	2	1	1	1	1	1	1	1	1	2	0	1	0	1	0	0	0	0
Animal activity	2	0	1	0	0	0	0	0	0	3	0	0	0	0	0	0	0	0
Total score	12	2	3	2	3	2	4	4	4	9	0	1	0	1	0	1	0	0

D—DIM Epi; M—Molnupiravir; 50/100/200—doses (mg/kg/day).

**Table 3 viruses-17-00964-t003:** Visual assessment of lung tissue lesions in hamsters infected with SARS-CoV-2 on day 3 post-infection; n = 3 (foci of bronchopneumonia are marked with a red arrow). The volume of hemorrhagic drainage pneumonia was independently assessed by two investigators and presented as a percentage.

	Viral Control	DIM Epi 50 mg/kg/day	DIM Epi 100 mg/kg/day	Molnupiravir 100 mg/kg/day	Molnupiravir 200 mg/kg/day	All Combination
Volume of hemorrhagic drain pneumonia	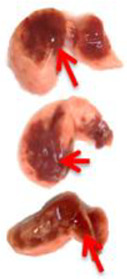	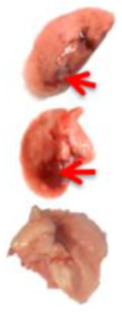	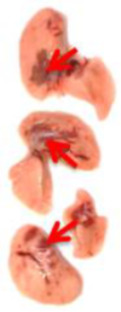	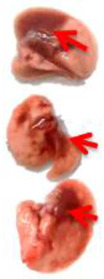	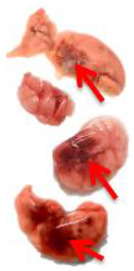	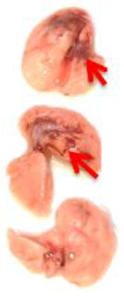
	60–70%	10–15%	10–15%	20–25%	15–20%	0–10%
Location of the lesion	Widespread in all lobes	Single; central zones	Single; central zones	Infrequently; central and regional zones	Infrequently; central, rarely regional zones	Single; central zones
Destruction of lung tissues	+	−	−	−	−	−

## Data Availability

The data presented in this study are available upon demand from the corresponding author.

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
