# Peer review of "3,3′-Diindolylmethane Improves the Viral Pneumonia Outcomes After Influenza and SARS-CoV-2 Infection in Animal Models"

_viruses, 2025, doi:10.3390/v17070964_

Round 1
Reviewer 1 Report
Comments and Suggestions for Authors
This study is overall fair and contains some interesting elements. However, it raises several serious concerns:
The authors mention ongoing clinical trials of DIM and reference multiple published studies on DIM and its derivatives against SARS-CoV-2 (DOI: 10.1002/jmr.3101) and influenza virus (DOI: 10.1016/j.jnutbio.2007.05.004). Given this background, the rationale for revalidating DIM in animals is unclear. What is the novelty or added value of this study? Moreover, there is no mention of animal ethics approval. Based on the current experimental design, I would be very surprised if this protocol had received IACUC approval. If the authors cannot provide valid ethical approval, or if none was obtained, that raises a serious issue.
The presented data are rather rough and incomplete. For example, clinical scores were not monitored or presented daily but shown only at two time points as individual values, which should instead be summarized in a figure or table for clarity. Additionally, the so-called “visual assessment” of lung lesions in Table 3 is not acceptable in its current form. There is no indication of where the hemorrhagic or pneumonic lesions were located, nor any comparative photos of affected versus control lungs. Quantitative metrics (e.g., 10%, 20%, 25%) are listed without explanation—what were the criteria for assigning these percentages?
Furthermore, the histological images in Figure 6 provide minimal insight. As a reader, I could not discern any meaningful differences between the groups, nor were any lesion characteristics or pathological changes clearly described.
In light of the issues above, I recommend rejection.
Author Response
We greatly appreciate the thoughtful comments, part of then are fair and very useful for us.
Comments 1: The authors mention ongoing clinical trials of DIM and reference multiple published studies on DIM and its derivatives against SARS-CoV-2 (DOI: 10.1002/jmr.3101) and influenza virus (DOI: 10.1016/j.jnutbio.2007.05.004). Given this background, the rationale for revalidating DIM in animals is unclear. What is the novelty or added value of this study? Moreover, there is no mention of animal ethics approval. Based on the current experimental design, I would be very surprised if this protocol had received IACUC approval. If the authors cannot provide valid ethical approval, or if none was obtained, that raises a serious issue.
Response 1: First of all we would like to point the novelty of our work. In our work, we studied a new composition Cesarox Epi, which differs from DIM and represents its modification with improved properties. We detailed the relevance of our work in the introduction.
The article (DOI: 10.1002/jmr.3101) investigates the interaction mechanisms between diindolylmethane derivatives and the main protease of SARS-CoV-2 (Mpro), identifying compound 1p as the most effective inhibitor and providing insights for the design of potential antiviral agents; however, it demonstrates only potential activity in vitro and does not include in vivo efficacy data or animal studies.
In addition, in the references (DOI: 10.1002/jmr.3101, DOI: 10.1016/j.jnutbio.2007.05.004) you indicated, the effect of DIM on immunological markers of infections was studied. However, these studies do not provide information on the therapeutic effects of the drug specifically on clinical and viral markers of infection (survival, weight loss, clinical signs, lung condition assessments). We consider it necessary to evaluate the positive effect of the new formulation specifically in generally accepted animal models of infection. We thank you for this comment, noted it in the Discussion and included the references you mentioned in the list of references.
\We apologize, that we missed information about ethical approval. All our studies were conducted according to “Approval of the Rules of Good Laboratory Practice” (Order of the Ministry of Health of the Russian Federation No. 199n dated April 1, 2016 ) and to the requirements of the European Convention for the Protection of Vertebrate Animals Used by the Institute for Experimental and Other Scientific purposes (Strasbourg, 18th March1986). The research protocol was approved by the Research Ethics Review Committee of the I. I. Mechnikov Research Institute for Vaccines and Sera, Moscow, Russia (approval â„– 12, 04.10.2023).We included information about this in the text of paper and enclose protocol of Approval and its translation in English for editors.
Comments 2: The presented data are rather rough and incomplete. For example, clinical scores were not monitored or presented daily but shown only at two time points as individual values, which should instead be summarized in a figure or table for clarity. Additionally, the so-called “visual assessment” of lung lesions in Table 3 is not acceptable in its current form. There is no indication of where the hemorrhagic or pneumonic lesions were located, nor any comparative photos of affected versus control lungs. Quantitative metrics (e.g., 10%, 20%, 25%) are listed without explanation—what were the criteria for assigning these percentages? Furthermore, the histological images in Figure 6 provide minimal insight. As a reader, I could not discern any meaningful differences between the groups, nor were any lesion characteristics or pathological changes clearly described.
Response 2: The reason that we included clinical scores at two time points ( day 3 and 7 after infection) was to estimate correlation the clinical signs with lung conditions that we assessed at these days. We agree that “visual assessment” of lung lesions in Table 3 is confusing. We have changed the Table 3, added information and photos about localization the hemorrhagic or pneumonic lesions, and listed criteria for assigning percentages of lesions. We also changed the histological images in Figure 6, where the arrows indicated the foci of pneumonia, to clearly illustrate any observed differences.

Reviewer 2 Report
Comments and Suggestions for Authors
The data in the figures nicely demonstrated the utility of DIM as an adjuvant therapy to standard antiviral drugs in influenza (in mice) and as a monotherapy in preventing pneumonia associated with SARS-CoV-2 (in Syrian hamsters) infection.
Please include the histology of the lungs of mice infected with influenza and treated with the monotherapy and combination therapy as was done with the hamsters to show the efficacy of the drug(s).
In the legend of Table 3, write when the effects were observed in days after infection.
Using temperature telemetry probably would be a more accurate measurement of temperature (fever) in hamsters (SARS-CoV-2)-there may be significant differences found over time with monotherapy and combination therapy. This should be commented on in the discussion if telemetry is not possible in the laboratory.
Author Response
Thank you for your insightful comments and your suggestions.
Comments 1: Please include the histology of the lungs of mice infected with influenza and treated with the monotherapy and combination therapy as was done with the hamsters to show the efficacy of the drug(s).
Response 1: We did not conduct histological study of lungs of mice, because the mouse influenza pneumonia is very convincing model of infection, causing total animal mortality in contrast to the non-lethal SARS-Cov2 infection model in hamsters. The survival of animals infected with influenza virus is the main and generally accepted criteria of the therapeutic effect of drugs. We agree that histological studies may also be important in this case, and we will conduct them in future studies whenever possible.
Comments 2: In the legend of Table 3, write when the effects were observed in days after infection.
Response 2: We have clarified this information in the revised manuscript
Comments 3: Using temperature telemetry probably would be a more accurate measurement of temperature (fever) in hamsters (SARS-CoV-2)-there may be significant differences found over time with monotherapy and combination therapy. This should be commented on in the discussion if telemetry is not possible in the laboratory.
Response 3: We fully agree with you that telemetry is very accurate measurement of temperature and probably will help to find more significant differences with monotherapy and combination therapy. Unfortunately we do not have this equipment in our Lab and this is one of the limitations of our study. According your suggestion we added comments on it in the Discussion.
Reviewer 3 Report
Comments and Suggestions for Authors
This manuscript investigates the efficacy of 3,3’-diindolylmethane (DIM) in mitigating the consequences of influenza virus infection in a mouse model and SARS-CoV-2 infection in a hamster model. The authors present evidence that DIM administration can reduce inflammation in vivo following infection and improve overall outcomes. Furthermore, the study explores the co-administration of DIM with established antiviral therapies: molnupiravir for SARS-CoV-2 and oseltamivir for influenza virus. Interestingly, the combination of DIM and oseltamivir demonstrated enhanced effectiveness against influenza, whereas co-administration of DIM with molnupiravir did not yield better results compared to treated alone.
Overall, the experimental design appears reasonable, and the data presented largely support the authors' conclusions regarding DIM's effectiveness. I have several suggestions for improving the clarity, accuracy, and completeness of the manuscript.
- The manuscript extensively discusses the "cytokine storm" and inflammation, yet it lacks a direct assessment of host immune responses. It is strongly suggested that the authors at least include a qRT-PCR panel to assess changes in key host immune-related gene expression (e.g., pro-inflammatory and anti-inflammatory cytokines, chemokines) to provide a more comprehensive understanding of DIM's immunomodulatory effects.
- The information regarding COVID-19 in the introduction appears to be outdated, referencing 2023 data in what is now 2025. Please update this section to reflect the current understanding and epidemiological data.
- Line 67-70: The description only mentioned the role of macrophages as the sole source of inflammatory mediators during cytokine storms in influenza and COVID-19. While macrophages are indeed crucial contributors, it is important to acknowledge that other immune cells, such as neutrophils, dendritic cells, and T cells, also play significant roles in cytokine production. Additionally, injured epithelial and endothelial cells can contribute to the release of cytokines. The pathogenesis of viral pneumonia and subsequent acute respiratory distress syndrome (ARDS) is multifactorial, encompassing direct viral damage, the specific viral strain, and the host's individual immune response, in addition to the cytokine storm. Therefore, it is inaccurate to attribute the cytokine storm solely to macrophage cytokine secretion. Accordingly, how DIM influences immune responses should be discussed (see suggestion 4). This section should be rephrased to reflect the broader cellular contributions, and a more comprehensive set of references should be provided.
- The authors used a commercially available DIM product (Cesarox@Epi). While it is appropriate to specify the product tested, the discussion should clarify that DIM is a naturally occurring compound and that research on its broader therapeutic potential extends beyond this specific commercial formulation. The manuscript should avoid generalizations about "all DIM" based solely on observations with the company's product. Furthermore, the discussion section currently contains two paragraphs that largely repeat the conclusions. This section could be significantly strengthened by expanding on the potential mechanisms by which DIM exerts its immunomodulatory effects (e.g., specific signaling pathways), discussing the potential for clinical translation (e.g., formulation, human dosage and application), and providing a more comprehensive understanding of its broader biological activities. For instance, including data or a discussion on the physiological effects of DIM at the tested dose in uninfected animals (e.g., lung morphology) would provide valuable background and context.
- There are several places of statistical significance that need clarification. For example, in lines 285-288, the text states that differences were not statistically significant, yet the corresponding figures appear to show significant differences between treated and control groups. It is likely the authors intended to state that there was no significant difference between the treated groups, but this needs to be clarified.
- Figure legends need some clarification. 1). The number of animals (n) used for each experimental group is missing from all figure legends. Please include "n=" for each experiment in the respective figure legends to indicate sample size. 2). The resolution of some figures, like Figure 2D, Figure 3D, and Figure 6, is not optimal and makes it difficult to discern details. Please improve the resolution of these figures. In Figures 2D and 3D, the X-axis labels are unclear. It is presumed the last bar represents "Viral control," but this is illegible. Please ensure all labels are clear and readable. The resolution of Figure 6 is very low, making it difficult to assess lung morphology. Please improve the resolution and consider including magnified insets or zoom views of specific lung regions to clearly illustrate any observed differences. 3). The order of bars representing viral control is inconsistent across figures. In Figures 2 and 3, the viral control is presented as the last bar, whereas in Figure 5, it is the first. Please maintain a consistent order for all comparable data presentations, preferably by placing the viral control first for logical flow. 4). The time points at which the results were obtained are not indicated in the figure legends. Please include the specific points for all data presented in the figures.
- Several sections of the manuscript indicate that data from multiple time points were collected for infected animals, but only one time point is presented in the figures or tables. For example, lines 471-474 mention Day 7 hamster data in Table 3, but this data is not consistently shown in Table3. Please ensure that all mentioned data points are presented or adequately explained.
Author Response
Thank you for your insightful comments and your suggestions.
Comments 1: The manuscript extensively discusses the "cytokine storm" and inflammation, yet it lacks a direct assessment of host immune responses. It is strongly suggested that the authors at least include a qRT-PCR panel to assess changes in key host immune-related gene expression (e.g., pro-inflammatory and anti-inflammatory cytokines, chemokines) to provide a more comprehensive understanding of DIM's immunomodulatory effects.
Response 1: We appreciate the reviewer’s valuable suggestion regarding the inclusion of qRT-PCR data to directly assess host immune responses. However, we were not able to perform gene expression analyses within the scope of the current study. Instead, we provided a detailed literature-based overview of the known immunomodulatory effects of DIM, including its impact on key inflammatory pathways and immune cell populations, as supported by multiple in vitro and in vivo studies. This overview was incorporated to offer a broader immunological context and to highlight relevant mechanisms of action while remaining within the feasible scope and format of the manuscript. We fully agree that future studies should include direct molecular analyses to further validate and expand upon the immunological effects of DIM.
Comments 2: The information regarding COVID-19 in the introduction appears to be outdated, referencing 2023 data in what is now 2025. Please update this section to reflect the current understanding and epidemiological data.
Response 2: We have updated information according new epidemiological data
Comments 3: Line 67-70: The description only mentioned the role of macrophages as the sole source of inflammatory mediators during cytokine storms in influenza and COVID-19. While macrophages are indeed crucial contributors, it is important to acknowledge that other immune cells, such as neutrophils, dendritic cells, and T cells, also play significant roles in cytokine production. Additionally, injured epithelial and endothelial cells can contribute to the release of cytokines. The pathogenesis of viral pneumonia and subsequent acute respiratory distress syndrome (ARDS) is multifactorial, encompassing direct viral damage, the specific viral strain, and the host's individual immune response, in addition to the cytokine storm. Therefore, it is inaccurate to attribute the cytokine storm solely to macrophage cytokine secretion. Accordingly, how DIM influences immune responses should be discussed (see suggestion 4). This section should be rephrased to reflect the broader cellular contributions, and a more comprehensive set of references should be provided.
Response 3: We have revised the text to acknowledge the broader range of immune and non-immune cells involved in cytokine production during viral pneumonia, including neutrophils, dendritic cells, T cells, and damaged epithelial and endothelial cells. We also clarified that the pathogenesis of ARDS is multifactorial and not solely dependent on macrophage-derived cytokines. Additional references have been included to support this more comprehensive view. We appreciate your suggestion and believe the revised section more accurately reflects the complexity of the inflammatory response in viral pneumonia.
We provide a more comprehensive understanding of DIM's in the Discussion section.
Comments 4: The authors used a commercially available DIM product (Cesarox@Epi). While it is appropriate to specify the product tested, the discussion should clarify that DIM is a naturally occurring compound and that research on its broader therapeutic potential extends beyond this specific commercial formulation. The manuscript should avoid generalizations about "all DIM" based solely on observations with the company's product. Furthermore, the discussion section currently contains two paragraphs that largely repeat the conclusions. This section could be significantly strengthened by expanding on the potential mechanisms by which DIM exerts its immunomodulatory effects (e.g., specific signaling pathways), discussing the potential for clinical translation (e.g., formulation, human dosage and application), and providing a more comprehensive understanding of its broader biological activities. For instance, including data or a discussion on the physiological effects of DIM at the tested dose in uninfected animals (e.g., lung morphology) would provide valuable background and context.
Response 4:Thank you so much for excellent suggestions. In response, we have revised the manuscript to clearly state that the study was conducted using a specific commercially available formulation, DIM Epi (Cesarox®Epi), and we now consistently refer to it as such throughout the text to avoid overgeneralization. We have emphasized that while DIM is a naturally occurring compound with broad therapeutic potential, the findings of this study are specific to the DIM Epi formulation used. Additionally, we have expanded the discussion to detail potential immunomodulatory mechanisms of DIM, including references to known signaling pathways involved in cytokine regulation. We have also added a more nuanced perspective on the translational potential of DIM, addressing considerations related to formulation, dosing, and clinical application. Furthermore, we clarified the physiological effects of DIM Epi at the tested dose in uninfected animals, including observations related to lung morphology, to provide appropriate context for interpreting the results. We have also revised the discussion to remove redundancy and improve clarity. We appreciate your suggestions, which have significantly strengthened the manuscript.
Comments 5: There are several places of statistical significance that need clarification. For example, in lines 285-288, the text states that differences were not statistically significant, yet the corresponding figures appear to show significant differences between treated and control groups. It is likely the authors intended to state that there was no significant difference between the treated groups, but this needs to be clarified.
Response 5: We apologize for inaccuracy and clarified statistical data.
Comments 6: Figure legends need some clarification. 1). The number of animals (n) used for each experimental group is missing from all figure legends. Please include "n=" for each experiment in the respective figure legends to indicate sample size. 2). The resolution of some figures, like Figure 2D, Figure 3D, and Figure 6, is not optimal and makes it difficult to discern details. Please improve the resolution of these figures. In Figures 2D and 3D, the X-axis labels are unclear. It is presumed the last bar represents "Viral control," but this is illegible. Please ensure all labels are clear and readable. The resolution of Figure 6 is very low, making it difficult to assess lung morphology. Please improve the resolution and consider including magnified insets or zoom views of specific lung regions to clearly illustrate any observed differences. 3). The order of bars representing viral control is inconsistent across figures. In Figures 2 and 3, the viral control is presented as the last bar, whereas in Figure 5, it is the first. Please maintain a consistent order for all comparable data presentations, preferably by placing the viral control first for logical flow. 4). The time points at which the results were obtained are not indicated in the figure legends. Please include the specific points for all data presented in the figures
Response 6: 1) We included number of animals in figure legends; 2) We have changed the resolution of Fig 2D, 3D, and 6, labeled X-axis on Fig 2D and 3D, and "Viral control," and highlighted specific lung regions to clearly illustrate observed differences. 3) We have changed Figures according to your remarks 4). We included the time points in the figure legends.
Comments 7: Several sections of the manuscript indicate that data from multiple time points were collected for infected animals, but only one time point is presented in the figures or tables. For example, lines 471-474 mention Day 7 hamster data in Table 3, but this data is not consistently shown in Table3. Please ensure that all mentioned data points are presented or adequately explained.
Response 7: We have not revealed obvious differences in the severity of inflammatory manifestations at this stage between the viral control group and the groups using mono- and combination therapy of both drugs in all studied doses on day 7 after infection. In order not to make the Figure very overloaded, we did not put this data on the drawings and described it in the text.
Reviewer 4 Report
Comments and Suggestions for Authors
In the manuscript by Kiselev et al., the author tested DIM( 3, 3’ Diindolymethane) treatment in mice with influenza (with or without Oseltamivir) and hamsters with SARS-CoV-2 (with or without Molnupiravir). DIM is a compound found in vegetables with potent to exhibit anticancer and anti-inflammatory properties. Molnupiravir is an antiviral medication inhibit certain RNA viruses replication. It is used to treat adults with mild to moderate COVID-19.
The authors treated tested animals with DIM, Oseltamivir/Molnupiravir or different combinations of the two. The authors used mice for influenza A(H1N1) infection and Syrian hamsters for SARS-CoV-2 infections. The authors evaluated the tested animals by death ratio, body weight, clinical signs, temperature, lung pathology, viral titer. The authors found 5-days treatment of DIM improved mice survival and prevented body weight lost, and combination were more effective. DIM in hamsters infected with SARS-CoV2 reduced clinical signs, weight loss, temperature elevation, and lung pathology. However, viral titer in animal lungs didn’t significantly change when DIM used alone. The authors used R-Studio and the “drc” package carried out the statistical analysis.
Overall, it is an interesting manuscript that showed DIM as a potent therapy for respiratory infections. The experiments and data were organized clearly in the tables and figures. However, before publication consideration, some minor revises may be required.
1, What does the authors mean at line 270 “beginning 4 hours before infection once daily”? Can the authors rewrite the whole sentence from 267 to 270 to make it more clear to readers?
2, line 146, which line does the author refer to by “’infection media’(detailed above)”?
3, Can the authors fix the “,” when “.” Is required? Otherwise, it is very confusing. For example,
a), line 41, “mately 7, 0 million…”
b), line 154, “4,5 g/L”
c), line 275, “exceeding >7,0 lg” “0,1ml”
d), line279, “by 1,7 times”
e), and others
4, Is there any typo error in line 54? “with 290.000 to 650.000 deaths globally[6].”
Author Response
We greatly appreciate the thoughtful comments and your suggestions. In response to their remarks, we have incorporated changes (point by point).
Comments 1: What does the authors mean at line 270 “beginning 4 hours before infection once daily”? Can the authors rewrite the whole sentence from 267 to 270 to make it more clear to readers?
Response 1: We apologize for confusing sentence and changed it to make it more clear
Comments 2: line 146, which line does the author refer to by “’infection media’ (detailed above)”?
Response 2: We apologize for inaccuracy and included full composition of media in text
Comments 3: Can the authors fix the “,” when “.” Is required? Otherwise, it is very confusing. For example,
a), line 41, “mately 7, 0 million…”
b), line 154, “4,5 g/L”
c), line 275, “exceeding >7,0 lg” “0,1ml”
d), line279, “by 1,7 times”
e), and others
Response 3: We apologize for inaccuracy in this part. We have made all corrections throughout the text
Comments 4: Is there any typo error in line 54? “with 290.000 to 650.000 deaths globally[6].”
Response 4: We regret the errors and have made the correction
Round 2
Reviewer 1 Report
Comments and Suggestions for Authors
-
The manuscript appears to be divided into two largely separate studies — one focusing on influenza and another on SARS‑CoV‑2. The authors used different animal models (mice vs. hamsters) and different FDA‑approved treatments (oseltamivir vs. molnupiravir), making it unclear why both studies were conducted in a single manuscript. Although both are respiratory viral infections, the rationale for combining these studies is not well justified, resulting in a disjointed presentation.
-
In the influenza study, the dosing for oseltamivir varied between experiments (20 mg/kg in the first study versus 5 and 10 mg/kg in the second). Including a 20 mg/kg group in the second experiment would have strengthened consistency and comparability, especially given that the first study appeared more thorough.
-
In the SARS‑CoV‑2 study, the choice of molnupiravir as the positive control is questionable, as other treatments like paxlovid have demonstrated stronger clinical efficacy. Moreover, the combined treatment of DIM and molnupiravir appeared to neutralize the protective effect of DIM, making this experimental design puzzling and diluting the conclusions.
-
Most critically, using different animal models (mice for influenza, hamsters for SARS‑CoV‑2) complicates direct comparisons and weakens the overall impact of the work. Hamsters can be used effectively for both infections and would also enable studies of co‑infection, making the results more coherent and impactful.
Minor Comments
-
No error bars are shown in Figure 5B. These should be added for clarity.
-
The images in Figure 6 require higher magnification and better quality for accurate assessment of the observed effects.
Author Response
We greatly appreciate the thoughtful comments, they are very useful for us and will be a trigger for further studies. What follows below is a point-by-point response to each comment raised by you
Comments 1. The manuscript appears to be divided into two largely separate studies — one focusing on influenza and another on SARS‑CoV‑2. The authors used different animal models (mice vs. hamsters) and different FDA‑approved treatments (oseltamivir vs. molnupiravir), making it unclear why both studies were conducted in a single manuscript. Although both are respiratory viral infections, the rationale for combining these studies is not well justified, resulting in a disjointed presentation.
RESPONSE 1: We thank the reviewer for this valuable comment. The rationale for combining the two models is justified in our intention to evaluate the universal potential of DIM Epi as a host-directed therapy for viral pneumonia, rather than to compare the effects of the drug on specific viruses. Influenza and SARS-CoV-2 are major respiratory pathogens that both cause pneumonia primarily through lung inflammation (doi: 10.1055/s-0043-1777796), which is often responsible for the severe course of the disease and which was the central focus of our study. Using available animal models, we aimed to assess whether DIM Epi could mitigate pneumonia-associated pathology across different viral infections. To improve clarity, we have revised the aim of the study in the Introduction section to better reflect this unified focus.
Comments 2: In the influenza study, the dosing for oseltamivir varied between experiments (20 mg/kg in the first study versus 5 and 10 mg/kg in the second). Including a 20 mg/kg group in the second experiment would have strengthened consistency and comparability, especially given that the first study appeared more thorough.
RESPONSE 2: We thank the reviewer for this comment and the opportunity to clarify our rationale. The 20 mg/kg dose of oseltamivir is a well-established and widely recommended in murine models of influenza (e.g., DOI: 10.1128/AAC.01579-07). In our initial experiment, this dose was highly effective, providing 75% survival and preventing body weight loss. Moreover, it significantly suppressed viral replication in the lungs (>2-log reduction compared to untreated controls: 3.42 ±â€¯0.38 vs. >7.0 log TCIDâ‚…â‚€/0.1 ml). In the second experiment, we intentionally used lower doses of both oseltamivir (5 and 10 mg/kg) and DIM Epi to better observe potential effects of combination therapy, which is a common approach in combination studies (e.g., DOI: 10.1055/a-1147-8824; DOI: 10.1177/095632020601700403).
Additionally, both experiments were conducted in parallel using animals from the same cohort and identical viral challenge doses, which allowed for reliable cross-comparison between treatment groups. We have now clarified this point in the Results section of the manuscript.
Comments 3: In the SARS‑CoV‑2 study, the choice of molnupiravir as the positive control is questionable, as other treatments like paxlovid have demonstrated stronger clinical efficacy. Moreover, the combined treatment of DIM and molnupiravir appeared to neutralize the protective effect of DIM, making this experimental design puzzling and diluting the conclusions.
RESPONSE 3: Thank you so much for interesting suggestion. Both molnupiravir and nirmatrelvir-ritonavir (Paxlovid) are currently licensed and recommended for treatment of COVID 19 infection 1. In our study, we selected molnupiravir as the positive control because it is more widely used in clinical practice in our country and was more available for preclinical research. Molnupiravir has demonstrated antiviral efficacy in multiple animal models of SARS-CoV-2 infection and was considered a suitable reference compound at the time of study design. However, we acknowledge that recent clinical data and meta-analyses suggest that the therapeutic effect of molnupiravir may be more limited than initially expected (e.g., Viruses, 2023; doi: 10.3390/v15112151), and that Paxlovid may represent a more robust clinical comparator.
Regarding the combination experiment, we agree that the lack of additive or synergistic benefit observed with DIM Epi and molnupiravir was unexpected. This may reflect a ceiling effect of antiviral treatment in the model, pharmacodynamic interactions, or limitations of the experimental design. We address this issue as one of the limitations of our study in the Discussion.
1World Health Organization. Therapeutics and COVID-19: living guideline. Geneva: WHO; 2023. Available from: https://www.who.int/publications/i/item/WHO-2019-nCoV-therapeutics-2023.2
Comments 4: Most critically, using different animal models (mice for influenza, hamsters for SARS‑CoV‑2) complicates direct comparisons and weakens the overall impact of the work. Hamsters can be used effectively for both infections and would also enable studies of co‑infection, making the results more coherent and impactful.
RESPONSE 4: We thank the reviewer for this important comment. Our primary goal was to demonstrate the efficacy of DIM Epi against both influenza and SARS-CoV-2 infections using reliable and widely accepted animal models for each virus. While Syrian golden hamsters use to study influenza virus pathogenesis and transmission, they typically do not exhibit overt clinical signs such as weight loss or lethality, which limits their use for evaluating antiviral efficacy1. In contrast, murine influenza pneumonia are well-established, peracute, and lethal models of influenza infection, commonly employed to assess antiviral activity.
Although lethal SARS-CoV-2 infection models in transgenic mice exist, they were not available to us at the time of study. Therefore, we selected the Syrian hamster model, which is widely accepted for preclinical evaluation of SARS-CoV-2 treatments due to its robust clinical disease phenotype.
We recognize the value of using hamsters for both infections and agree that model would provide more coherent and impactful results. We are currently developing a hamster model of influenza and SARS-CoV-2 coinfection. We appreciate your very interesting point and we will study DIM epi efficacy in this model in the future studies.
1Bouvier, N.M.; Lowen, A.C. Animal Models for Influenza Virus Pathogenesis and Transmission. Viruses 2010, 2, 1530-1563. https://doi.org/10.3390/v20801530
Minor Comments
Comments 1: No error bars are shown in Figure 5B. These should be added for clarity.
RESPONSE 1: We appreciate the comment and have added error bars to Figure 5B for clarity
Comments 2: The images in Figure 6 require higher magnification and better quality for accurate assessment of the observed effects.
RESPONSE 2: Thank you for your valuable suggestion. In the revised version of Figure 6, we have included higher-quality and improved images where possible, while remaining within the technical limits of the original material.
The reason to include low-magnification overview micrographs was to illustrate the level and distribution of pneumonia across lung tissue, which is the key differentiating feature between the groups. This justifies the use of broader field views in our histological analysis. We have also included high-magnification and highlighted the difference in the pathomorphological features in current version (circles outlines, Fig 6).

Reviewer 3 Report
Comments and Suggestions for Authors
In abstract line 18-19, Please replace (DIM Epi, Cesarox Epi) by (DIM). Here you are talking about a nature compound.
Figures 4B, 4C, and 5B appear to be missing error bars.
Author Response
We greatly appreciate the comments. What follows below is a point-by-point response to each comment raised by you.
Comments 1: In abstract line 18-19, Please replace (DIM Epi, Cesarox Epi) by (DIM). Here you are talking about a nature compound.
RESPONSE 1: We apologize for inaccuracy and changed this mistake
Comments 2: Figures 4B, 4C, and 5B appear to be missing error bars.
RESPONSE 2: We apologize for inaccuracy and added error bars